# Formation of Nanoporous Mixed Aluminum-Iron Oxides by Self-Organized Anodizing of FeAl_3_ Intermetallic Alloy

**DOI:** 10.3390/ma12142299

**Published:** 2019-07-18

**Authors:** Paulina Chilimoniuk, Marta Michalska-Domańska, Tomasz Czujko

**Affiliations:** 1Departmentof Advanced Materials and Technologies, Faculty of Advanced Technology and Chemistry, Military University of Technology, Kaliskiego 2 Street, 00-908 Warszawa, Poland; 2Institute of Optoelectronics, Military University of Technology, Kaliskiego 2 Street, 00-908 Warszawa, Poland

**Keywords:** anodization, nanopores, oxides, self-organization, FeAl_3_, intermetallic alloys

## Abstract

Nanostructured anodic oxide layers on an FeAl_3_ intermetallic alloy were prepared by two-step anodization in 20 wt% H_2_SO_4_ at 0 °C. The voltage range was 10.0–22.5 V with a step of 2.5 V. The structural and morphological characterizations of the received anodic oxide layers were performed by field emission scanning electron microscopy (FE-SEM). Therefore, the formed anodic oxide was found to be highly porous with a high surface area, as indicated by the FE-SEM studies. It has been shown that the morphology of fabricated nanoporous oxide layers is strongly affected by the anodization potential. The oxide growth rate first increased slowly (from 0.010 μm/s for 10 V to 0.02 μm/s for 15 V) and then very rapidly (from 0.04 μm/s for 17.5 V up to 0.13 μm/s for 22.5 V). The same trend was observed for the change in the oxide thickness. Moreover, for all investigated anodizing voltages, the structural features of the anodic oxide layers, such as the pore diameter and interpore distance, increased with increasing anodizing potential. The obtained anodic oxide layer was identified as a crystalline FeAl_2_O_4_, Fe_2_O_3_ and Al_2_O_3_ oxide mixture.

## 1. Introduction

Anodic aluminum oxide (AAO) allows the creation of highly ordered, hexagonally-arranged arrays of pores, which can be used as a template for further research to create new nanomaterials. Morphological features of formed anodic aluminum oxide are characterized by given parameters such as pore diameter (Dp), interpore distance (cell diameter; Dc) and oxide layer thickness. Geometrical parameters are strongly affected by the applied voltage and temperature of the electrolyte. The influence of experimental conditions on AAO morphology also includes changes in porosity and pore density [1]. The best hexagonal pore arrangement can be achieved via self-organized anodization only in a very narrow range of applied potentials, known as the self-ordering regime, which is strongly dependent on the kind of electrolyte used. To obtain a variety of aluminum oxide morphologies, electrolytes such as sulfuric acid [2], oxalic acid [3,4] and phosphoric acid are usually applied [4]. Nevertheless, the most popular electrolyte used for anodization aluminum is sulfuric acid. For a typical mild anodization carried out in sulfuric acid, the voltage is approximately 25 V. However, the previously mentioned correlation between the experimental parameters and obtained morphology was also obtained during the anodization of metals such as Co [5], V [6] and intermetallic-based alloys, such as Ni_3_Al, TiAl [7,8] and FeAl [9,10].Tsuchiya et al. [7] reported the anodization of TiAl intermetallic alloy conducted in 1 M sulfuric acid with little addition of fluoride anions at various voltages (10 V, 20 V and 40 V). As a result, oxides with pore diameters from 38 nm to 100 nm were obtained. Stępniowski et al. [10] described the anodization of FeAl intermetallic alloy carried out in 20 wt% H_2_SO_4_ at 0 °C for 60 s at voltages ranging from 5 V to 20 V with a step of 2.5 V. It was found that the obtained morphology was characterized by a highly porous structure with small pores reaching up to approximately 30 nm at the highest applied voltage [10].However, in one of our previous papers, we presented a process that allows us to obtain an ultrasmall porous “spongy” structure by anodizing the FeAl intermetallic alloy using an electrolyte consisting of a 0.3 M solution of oxalic acid with the addition of 20% glycol [11].

The oxides obtained on intermetallic phases after oxidation have an amorphous structure. For example, after the anodization of the FeAl-based intermetallic alloy [10], an amorphous nanometric film is formed. After annealing, the anodic oxide is transformed into a mixture of crystalline spinel FeAl_2_O_4_ and Al_2_O_3_ phases. Additionally, the anodization of the Ni_3_Al phase leads to the formation of an amorphous oxide coating [12,13]. According to the stoichiometric chemical composition of the FeAl and FeAl_3_ phases, the contents of aluminum are 50 at% and 75 at%, respectively. The concentration of Al in the substrate material might affect the anodization process and influence the structure and phase composition of the obtained oxides.

The main goal of this paper is to demonstrate whether the anodization of a FeAl_3_ intermetallic alloy in sulfuric acid allows for the formation of anodic oxide layers with nanoporous morphology. In this paper, for the first time, we report on the possibility of a direct formation of crystalline oxides at a low anodization voltage of 10.0–22.5 V. Moreover, the correlation between the anodization parameters and morphology as well as the phase composition of the obtained oxides is presented.

## 2. Materials and Methods

The FeAl_3_ intermetallic alloy was cut into coupons with 0.9 mm thickness. The chemical composition of the FeAl_3_ intermetallic alloy was found to be as follows: 22.47 at% of Fe and 77.53 at% of Al. Prior to anodization, samples were degreased (acetone and ethanol) and electrochemically polished in a solution of HNO_3_ in ethanol (3:1 volumetric ratio). Electropolishing was carried out at the potential of 15 V at temperature −5 °C and time 300 s. The system for carrying out the anodization process consisted of a double-walled electrochemical cell with a water jacket. To ensure a constant temperature of the electrolyte during anodization, electrochemical cells located on a magnetic stirrer were connected to a circulator and a thermostat. The anodizing process was carried out in a two-electrode system. As a cathode, a 1 mm thick platinum electrode with a working surface of 900 mm^2^ was used. The anode was the FeAl_3_ base material with a working area of 20 mm^2^. Anodization was carried out in 20 wt% H_2_SO_4_ in the voltage range of 10.0–22.5 V with a step of 2.5 V at 0 °C. After one minute of the first step of anodization, a poorly ordered oxide was removed by chemical etching for five minutes in a vigorously stirred mixture of 6 wt% H_3_PO_4_ and 1.8 wt% H_2_CrO_4_ at 60 °C. After oxide removal, the reanodization was conducted under the same set of experimental conditions as the first step (one-minute experiments).

The morphological characterization was performed by field emission scanning electron microscopy (FE-SEM, Quanta 3D FE-SEM, FEI, Hillsboro, OR, USA). For each sample, three images at the same magnification (100,000×) were taken to evaluate the morphological features of the formed anodic oxide. Pore diameters and pore densities were estimated with the use of ImageJ, while the interpore distance was calculated from a fast Fourier transform (FFT) of the FE-SEM images using WSxM 5.0 Develop 6.2. X-ray diffraction patterns were recorded by XRD Rigaku Ultima IV (Neu-Isenburg, Germany) using CoK_α_ radiation in the range of 2Θ = 30–90° with a step of 0.01° and an acquisition rate of 1°/min. Crystallographic databases (PDF-2 2003, PDF-4 + 2014) and PDXL 2.1 were used to identify the phase composition.

## 3. Results and Discussion

Figure 1 shows the current density-time transients during the second anodization of the FeAl_3_ intermetallic alloy. In the initial stage of anodization, there was a sudden decrease in the current density and then its stabilization. This indicated the formation of boundary layer and the formation of nanoporous oxide coating. The course of the current curves were similar to the curves obtained during aluminum anodizing [14,15]. However, with the same set of operating conditions, the current density observed for FeAl_3_ anodizing was more than one order of magnitude higher than for pure aluminum [16] and 20 times less than for FeAl [10]. This was probably due to the iron content in the substrate material and the associated ion density in the iron in the electrolyte.

A representative morphology of the anodized oxide formed on the FeAl_3_ intermetallic alloy obtained at various voltages is shown in Figure 2. For every experimental condition, a highly nanoporous structure of the oxide layer was formed. The anodizing potential significantly affected the morphology of the outer oxide layer. It is clearly visible that morphological parameters such as pore diameter and the distances between the pores increased with increasing voltage. The poorly arranged pores formed the domain structure related to the local current density fluctuation. A similar effect was observed during the anodization of an FeAl-based intermetallic alloy in sulfuric acid [9,10,11]. To confirm this quantitatively, the average pore diameter was estimated from the FE-SEM images. The pore diameter increased linearly with the applied potential. The pore diameter of the anodic oxide obtained at 10 V was 16 ± 3 nm, and it increased to 32 ± 4 nm at 22.5 V (Figure 3A).

Such small nanopores, which are comparable to the literature data for anodic aluminum oxide, are usually difficult to obtain [17,18]. The pore diameter for the anodic oxide formed at high voltage was much larger than that for the material obtained at low voltage, and the difference was up to 16 nm.

The pore diameters at a low voltage of 10 V obtained under similar anodization conditions (the type and temperature of electrolyte) for the anodic oxide formed on pure aluminum [10], FeAl_3_ and FeAl [10] were 12 nm, 15 nm and 25 nm, respectively. At higher voltages, the pore diameters were much larger for the anodic oxide formed on the FeAl_3_ intermetallic alloy than those of the anodic oxide formed on high-purity alumina in the same voltage range in sulfuric [17] and oxalic acids [2]. However, the pore diameters were smaller for the oxide formed on FeAl_3_ than those of the oxide formed on FeAl with the same voltage and electrolyte [10]. The pore diameters for anodic oxides obtained at 20 V in sulfuric acid on pure aluminum, FeAl_3_ and FeAl substrates are 20 nm, 32 nm and 60 nm, respectively.

Linear growth along with the anodizing potential could also be seen by analyzing the distances between the pores. Analogously, the interpore distance linearly increased with applied voltage from 41 ± 2 nm at 10 V to 60 ± 2 nm at 22.5 V (Figure 3B). The values of the interpore distance were greater than those estimated for high-purity aluminum at the same voltage for anodization carried out in sulfuric [17] and oxalic acids [18] and smaller than those for FeAl [10]. The interpore distances for anodic oxides manufactured on pure Al [10], FeAl_3_ and FeAl [10] at 10 V were 25 nm, 40 nm and 50 nm, respectively. The increase in the anodization voltage up to 20 V results in interpore distance increased to 45 nm, 52 nm and 110 nm for oxides obtained on pure Al, FeAl_3_ and FeAl, respectively. In addition, it was found, on the basis of the obtained results, that the values of the distance between the pores did not change in a wide range such as that during the anodization of another intermetallic phase from the same system, namely, the FeAl phase. Almost a nine-fold increase in the interpore distance was observed for the FeAl intermetallic alloy anodized at 20 V in comparison to the values recorded at the 5 V potential [10].

Nevertheless, a decrease in the density of pores (Figure 3C) on the surface of the anodic oxide was observed along with the increase in voltage. The pores density ranges from 480 pores/μm^2^ at a potential of 10 V to 247 pores/μm^2^ at 22.5 V.The value at the lowest voltage was approximately twice as high as that at 22.5 V. The interpore distance is strongly related to the pores density [2,18]. As a consequence of the larger interpore distance for the anodic oxide formed on FeAl_3_ relative to that formed on Al, fewer pores were found on a given surface area. Thus, the pores density for the oxide formed on FeAl_3_ was much lower than that formed on anodic alumina in the same voltage range. Figure 4 shows the cross-sectional FE-SEM images of the anodic oxide formed in 20 wt% H_2_SO_4_.

On the basis of the obtained cross-sections, a pronounced increase in the thickness of the oxide is noted along with the increase in the anodizing potential. The thickness ranged from 0.61 μm for low voltage (10 V) to 7.74 μm for high voltage (22.5 V). To obtain a nanoporous structure with a precisely controlled oxide layer thickness, the adjustment of the anodizing duration is a very important issue.

In general, the rate of oxide growth depends mainly on the type of electrolyte, concentration, anodizing voltage, temperature of the process, and the type of substrate material. In particular, the oxide growth rate on the FeAl_3_ intermetallic alloy gradually increased up to 464 μm/h at 22.5 V. For comparison, the growth rates for Al and FeAl under similar process conditions were 56 μm/h and 743 μm/h, respectively [10].

Based on the observation of changes in the morphological parameters and the rate of growth of oxides produced under similar conditions, it can be concluded that with increasing iron content in the substrate, the pore diameter, the interpore distance and the growth rate of the oxide film increase.

Additionally, an X-ray phase analysis was carried out to identify the phase-produced anodic nanoporous oxide coating. The X-ray diffraction (XRD) patterns, shown in Figure 5, reveal that no amorphous phase was present in the structure of the anodic oxide. In the diffraction patterns, one can distinguish many peaks originating from the crystalline FeAl_2_O_4,_ Fe_2_O_3_ and Al_2_O_3_ phases. Considering the low anodization voltage, the lack of an amorphous form of aluminum oxide, which is typical for aluminum anodization [19], was unusual. However, at high current densities, one would expect a plasma electrolytic oxidation mechanism, which allows the formation of a crystalline oxide with the predominance of the Al_2_O_3_ phase [20,21]. Moreover, iron anodization leads also to the formation of an amorphous Fe_2_O_3_ oxide, which for crystallization is usually subjected to an annealing process. As a result of this process, crystalline anode iron oxide Fe_2_O_3_ is obtained on the iron surface [22]. The formation of iron oxide in the amorphous form is typical for the anodization process and occurs even during the preparation of ultra-thin layers [23]. A similar effect in the formation of a mixture of oxides was observed during the anodization of an intermetallic FeAl phase [10]. However, in contrast to the FeAl_3_ phase, the oxide coating obtained on FeAl was characterized by an amorphous structure.

It should be noted that along with the increase in the anodization potential, the fraction of spinel phases and complexity of phase composition increases. Such a significant fraction of iron in the observed oxide phases may have a significant impact on the physical properties of the coatings, especially the value of the energy gap [10,11]. The quantitative analysis of the contribution of the individual oxides and their impact on the properties of the oxide nanostructures that are formed on the FeAl_3_ substrate is the subject of further investigation.

## 4. Conclusions

The results for the self-organizing anodization of an FeAl_3_ intermetallic alloy lead to the following conclusions: The self-organized two-step anodization of an FeAl_3_ intermetallic alloy allows the formation of porous oxide in 20 wt% sulfuric acid at voltages ranging from 10 to 22.5 V. The pore diameter and interpore distance of the obtained nanoporous oxide increases linearly with the voltage as is observed during the anodization of pure aluminum or FeAl alloy. The pore density for the oxide formed on the FeAl_3_ intermetallic alloy was much lower than that of the anodic oxide formed on aluminum. The anodic oxide that formed on the FeAl_3_ intermetallic alloy had a high growth rate of up to 464 μm/h at the maximum applied voltage. Finally, the anodic oxide layer consisted of a crystalline FeAl_2_O_4_, Fe_2_O_3_ and Al_2_O_3_ oxide mixture.

## Figures and Tables

**Figure 1 materials-12-02299-f001:**
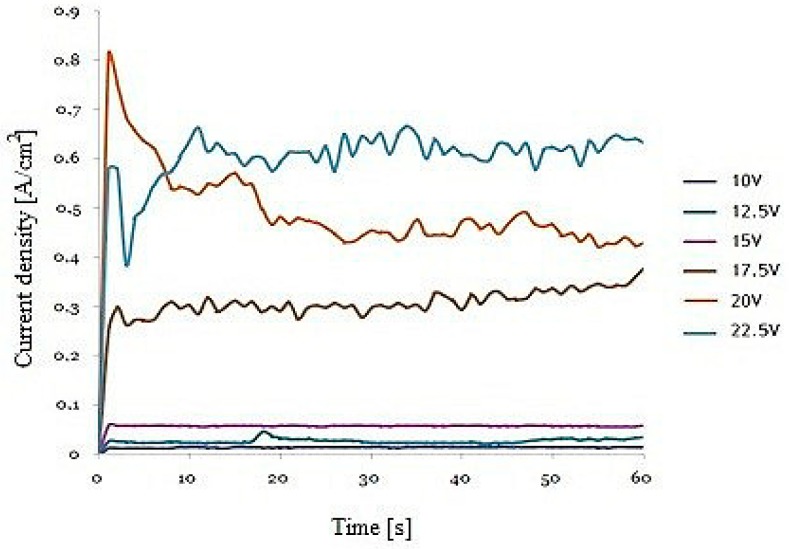
Current density vs. time for the second step of FeAl_3_ self-organized anodization in 20 wt% H_2_SO_4_ at 0 °C.

**Figure 2 materials-12-02299-f002:**
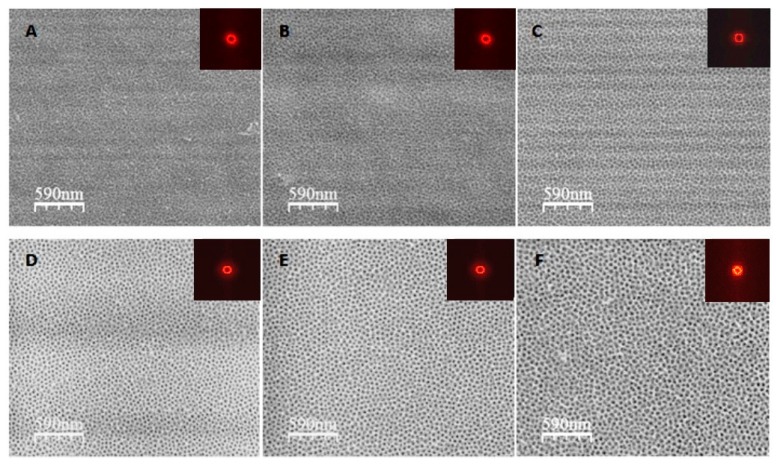
Top-viewfield emission scanning electron microscopy (FE-SEM) images and fast Fourier transform (FFT; right hand upper inserts) of nanoporous oxide formed by two-step anodization of FeAl_3_ intermetallic alloy at voltages in the range of 10–22.5 V with a step of 2.5 V (**A**–**F**).

**Figure 3 materials-12-02299-f003:**
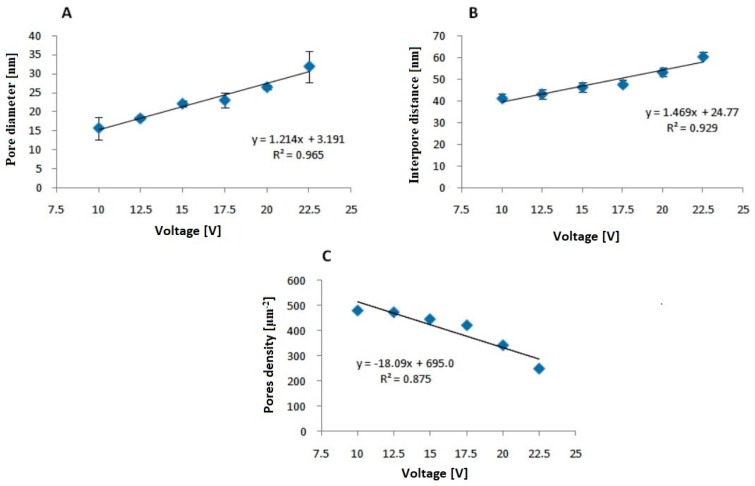
Pore diameter (**A**) interpore distance (**B**) and pore density (**C**) as a function of anodizing voltage.

**Figure 4 materials-12-02299-f004:**
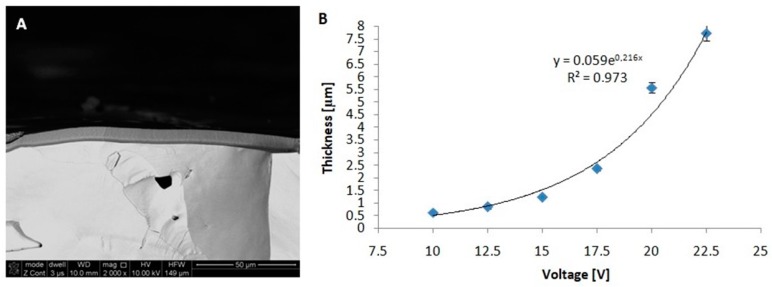
Cross-sectional FE-SEM image of the anodic oxide formed by self-organized anodization of the FeAl_3_ intermetallic alloy in 20 wt% H_2_SO_4_ at 22.5 V (**A**) and the thickness of the oxide layer as a function of the anodizing voltage (**B**).

**Figure 5 materials-12-02299-f005:**
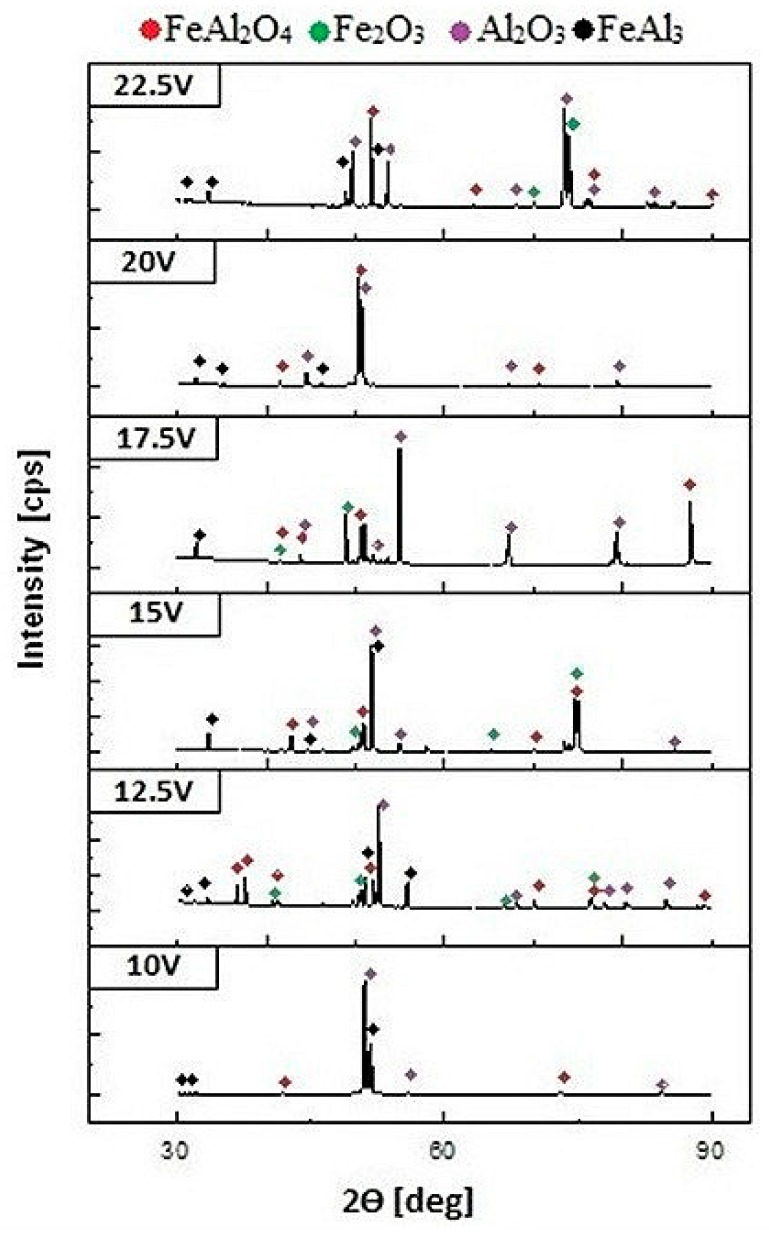
X-ray diffraction (XRD) patterns of the anodic oxide formed via self-organized two-step anodization of an FeAl_3_ intermetallic alloy.

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
