# Peer review of "Formation of Nanoporous Mixed Aluminum-Iron Oxides by Self-Organized Anodizing of FeAl3 Intermetallic Alloy"

_materials, 2019, doi:10.3390/ma12142299_

Reviewer 1 Report

The paper reports results on the fabrication of porous aluminum and iron oxides by anodizing of Fe3Al intermetallic alloy. Nanomorphology of the porous oxide is characterized quantitatively by SEM observations. Interestingly, the authors found that the porous oxide formed by anodizing Fe3Al intermetallic alloy consisted of crystalline FeAl2O4, Fe2O3, and Al2O3. This interesting result will contribute to the research field of anodizing science and technology. I have only three suggestions for the publication as follows:

1) Line 70 “electrochemically polished in a solution of HNO3in ethanol” Please describe the electrochemical conditions during the polishing of Fe3Al intermetallic alloy.

2) Line 156-162 If possible, please describe the crystallinity of the porous oxide formed by anodizing of pure Fe using references. I think several research groups have reported it.

3) Correct Ref. 17. “Nanoporous aluminum” to “Nanoporous alumina”, “2019” may be “2015”.

Reviewer 2 Report

The subject of the manuscript is interesting and adds significant contribution to the field of anodic films. I recommend the publication of the article after few clarifications from the authors. 

In order to understand the nanostructuring processes of these new systems, based on metal alloys. The analysis of current/time curves is fundamental. The authors must include in the manuscript the analysis of current/time transients generated during the anodization process.

In the materials and methods section, the technique used is not described in detail. For example, electrochemical arrangement, cell type, electrodes used, etc. Only used solutions and voltages are described.

Reviewer 3 Report

Some typing errors (superscripts and subscripts) need to be corrected (lines 80 and 81). Otherwise, I agree with this version of manuscript, and I think that it can be accepted after revision.

Author Response

Thank you for your valuable comments. We have corrected the superscripts and subscripts in the manuscript.